# The Infratentorial Localization of Brain Metastases May Correlate with Specific Clinical Characteristics and Portend Worse Outcomes Based on Voxel-Wise Mapping

**DOI:** 10.3390/cancers13020324

**Published:** 2021-01-17

**Authors:** Zhangqi Dou, Jiawei Wu, Hemmings Wu, Qian Yu, Feng Yan, Biao Jiang, Baizhou Li, Jinghong Xu, Qi Xie, Chenguang Li, Chongran Sun, Gao Chen

**Affiliations:** 1School of Medicine, Zhejiang University, Hangzhou 310058, Zhejiang, China; douzhangqi@zju.edu.cn (Z.D.); 21818268@zju.edu.cn (J.W.); hemmings@zju.edu.cn (H.W.); yuqian950627@zju.edu.cn (Q.Y.); fengyanzju@zju.edu.cn (F.Y.); jiangbiao@zju.edu.cn (B.J.); alexlibz@126.com (B.L.); zydxjh@zju.edu.cn (J.X.); 2315084@zju.edu.cn (C.L.); 2Department of Neurosurgery, Second Affiliated Hospital, School of Medicine, Zhejiang University, Hangzhou 310009, Zhejiang, China; 3Department of Radiology, Second Affiliated Hospital, School of Medicine, Zhejiang University, Hangzhou 310009, Zhejiang, China; 4Department of Pathology, Second Affiliated Hospital, School of Medicine, Zhejiang University, Hangzhou 310009, Zhejiang, China; 5School of Life Science, Westlake University, Hangzhou 310024, Zhejiang, China; xieqi@westlake.edu.cn

**Keywords:** brain metastases, magnetic resonance imaging, infratentorial localization, prognosis, voxel-wise analysis

## Abstract

**Simple Summary:**

Brain metastases (BMs) are cancerous lesions that originated from cancers outside the brain. Specific types of BMs are found distributing in specific brain areas. The infratentorial regions are frequently affected, causing severe neurological symptoms. Thus, it is necessary to investigate what types of tumors tend to form infratentorial BMs and whether these lesions are more fatal. By analyzing substantial brain imaging data of BMs, we found the vulnerability of infratentorial regions to most types of BMs, and found the infratentorial localization of BMs was significantly associated with young age, male sex, lung neuroendocrine and squamous cell carcinomas, more active cell division of the tumors, and patients with poorer outcomes. Additionally, infratentorial involvement might predict worse outcomes for patients who received surgery. Our findings underlined the distinctive role of infratentorial localization of BMs and its potential relationship with specific clinical characteristics, which may assist diagnosis, treatment choice, and prognostic prediction of BMs.

**Abstract:**

The infratentorial regions are vulnerable to develop brain metastases (BMs). However, the associations between the infratentorial localization of BMs and clinical characteristics remained unclear. We retrospectively studied 1102 patients with 4365 BM lesions. Voxel-wise mapping of MRI was applied to construct the tumor frequency heatmaps after normalization and segmentation. The analysis of differential involvement (ADIFFI) was further used to obtain statistically significant clusters. Kaplan-Meier method and Cox regression were used to analyze the prognosis. The parietal, insular and left occipital lobes, and cerebellum were vulnerable to BMs with high relative metastatic risks. Infratentorial areas were site-specifically affected by the lung, breast, and colorectal cancer BMs, but inversely avoided by melanoma BMs. Significant infratentorial clusters were associated with young age, male sex, lung neuroendocrine and squamous cell carcinomas, high expression of Ki-67 of primaries and BMs, and patients with poorer prognosis. Inferior OS was observed in patients with ≥3 BMs and those who received whole-brain radiotherapy alone. Infratentorial involvement of BMs was an independent risk factor of poor prognosis for patients who received surgery (*p* = 0.023, hazard ratio = 1.473, 95% confidence interval = 1.055–2.058). The current study may add valuable clinical recognition of BMs and provide references for BMs diagnosis, treatment evaluation, and prognostic prediction.

## 1. Introduction

Brain metastases (BMs) are one of the major causes of cancer mortality. The incidence of BMs is reported to be 10 times higher than that of primary brain tumors. In addition, 7–14 individuals per 100,000 or 10–30% of patients with systemic cancer may be affected [1,2,3]. The resistance of the blood-brain barrier to systemic chemotherapy and approximately 40% of patients affected by multiple metastatic lesions make BMs a challenging disease with a dismal prognosis [4]. In addition to systemic chemotherapy, conventional managements including surgery, whole-brain radiotherapy (WBRT), and stereotactic radiosurgery (SRS) play an irreplaceable role when treating BMs [5].

Several BM characteristics such as number, size, location, and peritumoral edema are necessary to be evaluated when adapting therapies [6]. Therein, the spatial distribution of BMs, which largely impacts the symptoms of patients, is an important biological factor to determine the local therapy regimens, and, subsequently, relates to the prognosis [7,8]. Neuroimaging techniques, especially magnetic resonance imaging (MRI), are feasible to detect the BMs sites in a clinical routine. More importantly, analyses mapping the distribution of BMs were conducted using MRI voxel-based methods, and certain associations between BMs locations and primary malignancies were found. The preferred localization of BMs at the grey-white matter junction and the watershed regions had been well documented, which was explained by the trapping of tumor emboli in the terminal supply areas of large intracranial arteries [9,10,11]. Further investigations regarding voxel-based landscapes of BM sites similarly demonstrated that the posterior fossa was a predilection watershed site of BMs from lung and breast cancers [10,12,13,14,15]. However, melanoma BMs were described as presenting aversion to the cerebellum [15,16]. These results benefited differential diagnoses and raised the attention of applying prophylactic cranial irradiation (PCI) for high-risk regions to develop BMs [10,14].

Known as the infratentorial regions, the posterior fossa was non-uniformly affected by distinct primaries, as mentioned. However, voxel-wise descriptive analyses warrant further statistical investigations, and the associations between infratentorial localization of BMs and other clinical characteristics remain unknown. The prognostic value of infratentorial involvement of BMs was also controversial. The current study, therefore, retrospectively, investigated the potential relationship between infratentorial localization of BMs and clinical characteristics including age, sex, primaries, Ki-67 index, and overall survival (OS) by voxel-wise mapping, and analyzed the independent risk factor of prognosis.

## 2. Results

### 2.1. Demographics and MRI Data Processing

According to the inclusion and exclusion criteria, patients with BMs were selected via either histopathological and radiological reports of brain lesions (*n* = 222) or histopathological results of primary tumors and radiological results of brain lesions (*n* = 880). Therefore, a total of 1102 patients (667 males and 435 females) with an average age of 59.8 years were included (Figure 1A). The most common primary malignancies were lung cancer (86.1%), followed by breast cancer (4.2%), colorectal cancer (2.5%), head and neck cancer (1.3%), kidney cancer (1.1%), melanoma (0.6%), and other cancers (4.2%). The median OS time of patients with follow-up (*n* = 991) was 10.84 months. Treatment data were available in 402 patients, showing 48.3% received whole-brain radiotherapy (WBRT) alone and 36.3% received surgery alone. Demographics are summarized in Table 1. For MRI data processing, format conversion, normalization, and semi-automatic segmentation were successfully conducted to obtain the regions of interest (ROIs, Figure 1B).

### 2.2. The Number, Volume, and Relative Metastatic Risk of BMs

A total of 4365 BMs were semi-automatically segmented. Patients with 1, 2, and ≥3 BMs accounted for 49.4%, 17.0%, and 33.6%, respectively (Table 1). Lung cancer (specifically, lung adenocarcinoma) caused the greatest number of BMs (Figure 2A). The number, total volume (TV), and volume of the single lesion (VSL) of BMs were investigated according to different regions and primaries. The highest average number was observed in the frontal lobe, while relatively higher TV and VSL were displayed in the insular lobe (Appendix A). Though no statistically significant difference was found in the average number of BMs between primaries, the TV of lung cancer BMs was the smallest, while kidney and colorectal cancers, and melanoma BMs displayed larger VSL (Appendix A). Three subtypes of lung cancer including adenocarcinoma, squamous cell carcinoma, and small cell cancer were further compared. We found squamous cell carcinoma BMs presented a larger VSL, but occurred less frequently when compared to adenocarcinoma and small cell cancer BMs (Appendix A).

Moreover, when comparing between hemispheres, the VSL in the right insular lobe was significantly higher than that in the left (Figure 2B). We further calculated the relative metastatic risk (RMR) in different lobes to measure the likelihood of BMs occurrence. After normalizing the metastatic risk to the volume of the corresponding brain region and obtaining the relative metastatic risk of BMs in the per unit volume of a specific brain region, we found BMs’ susceptibility (RMR > 1) in the parietal, insular, and left occipital lobes and in the cerebellum. The frontal lobe, however, showed BMs non-susceptibility (RMR < 1). The temporal and right occipital lobes, limbic system, thalamus and basal ganglia, and the brainstem showed similar BMs’ non-susceptibility (RMR < 1, Figure 2B).

### 2.3. The Landscape of BMs Sites According to Different Primaries

The spatial distribution of the 4365 BMs centroids categorized by the primary malignancies was displayed (Appendix A). The infratentorial region was one of the areas frequently affected by BMs. The tumor frequency heatmap was constructed by ROIs overlapping (Figure 1B). The results illustrated the majority of BMs were located in the frontal, parietal, and occipital lobes, and in infratentorial regions (Figure 3). Remarkable localization of BMs in the grey-white junction and watershed regions could be observed, as well as mild left lateralization. When categorized by different primaries, lung cancer BMs exhibited the preferred sites similar to that of all BMs, due to its high proportion. Moreover, predominant clusters were observed in the right cerebellum for breast cancer BMs, supratentorial regions for melanoma BMs, cerebellar vermis for colorectal cancer BMs, left frontal lobe for kidney cancer BMs, bilateral frontal lobes for head and neck cancer BMs, and infratentorial regions for other cancer BMs (Figure 3).

### 2.4. ADIFFI Showing Association between Infratentorial Localization and Clinical Characteristics of BMs

The *p*-value heatmaps comparing two differential phenotypes under one clinical characteristic were constructed to calculate the significance of a particular voxel, based on the analysis of differential involvement (ADIFFI) [15,16]. The heatmap for patients stratified by age (median = 60.5 years) suggested a predominance in the supratentorial hemispheres among older patients and in the infratentorial areas and the left frontal lobe among younger patients (Fisher’s exact test, *p* < 0.05, Figure 4A). When stratified by sex, it showed a highly specific localization in the parafalx areas among female patients and in the right frontal lobe, left parietal lobe, and left cerebellum among male patients (Figure 4B).

According to the subtypes of lung cancer, results revealed that lung epithelial carcinoma BMs occurred more frequently in the right parietal and occipital lobes, while lung neuroendocrine carcinoma BMs occurred more frequently in the left temporal lobe and the infratentorial regions, especially in the left cerebellum (Figure 4C). BMs from lung adenocarcinoma and squamous cell carcinoma, which are the two major subtypes of lung epithelial carcinoma, were further compared. The results indicated a predominance in the cortical areas and the cerebellum for lung squamous cell carcinoma BMs (Figure 4D).

In addition, BMs from primary malignancies that expressed Ki-67 (a marker of tumor proliferation) ≥ the median level (35%) showed significant localization in the left parietal and occipital lobes and the infratentorial regions (Figure 4E). BMs with Ki-67 ≥ the median (45%) showed significant localization in the left cerebellum (Figure 4F).

### 2.5. The Potential Relationship between Infratentorial BMs and Poor Prognosis

The follow-up data were collected from 991 patients. The median survival time was 10.84 months (Table 1). Shorter OS time was found in patients with an age ≥ the median (60.71 years, *p* < 0.001) and male sex (*p* < 0.001, Appendix A). Patients with melanoma BMs showed decreased OS when comparing those with BMs originated from other primaries, though the results were not significant (*p* = 0.0945, Figure 5A). The OS of BM patients with subtypes of lung cancer was further analyzed, and results indicated a better prognosis in patients with lung epithelial carcinoma BMs compared to those with lung neuroendocrine carcinoma BMs (*p* < 0.001, Appendix A). More specifically, patients with lung adenocarcinoma BMs experienced a significantly better OS than patients with lung squamous cell carcinoma and small cell cancer BMs (*p* < 0.001, Appendix A). In addition, poorer OS was displayed in patients with a Ki-67 index in primary malignancies ≥ the median (35%, *p* = 0.0183), and a Ki-67 expression level in BMs ≥ the median (45%, *p* = 0.0484, Appendix A).

Having ≥3 BMs contributed to a remarkably worse OS compared to having 1 or 2 BMs (*p* = 0.0036, Figure 5B). Importantly, though no statistically significant difference was found between patients with or without infratentorial involvement of BMs, the prognosis of the former tended to worsen after approximately a one-year follow-up (*p* = 0.0673, Figure 5C).

Additionally, a significantly worse OS was observed in patients who only received WBRT, compared with those who received surgery, surgery plus WBRT, or surgery plus stereotactic radiosurgery (*p* = 0.0020, Figure 5D).

The ADIFFI was used to investigate the survival-related localization of BMs. The comparison of patients with different survival time (>1 year vs. ≤1 year) identified significant clusters mainly in the left frontal and parietal lobes and the right occipital lobes that were associated with longer survival time, while clusters in the bilateral frontal lobes and cerebellum were associated with shorter survival time (Figure 5E).

### 2.6. Infratentorial Involvement of BMs Was an Independent Risk Factor of Poor Prognosis for Patients Who Received Surgery

Surgery is commonly the top priority for treating patients with a single brain-metastatic lesion, and surgical removal is also recommended in selected patients with multiple or recurrent BMs [5]. As a result, we analyzed the prognostic risk factor of prognosis in 200 patients who received surgery in our cohort. Factors including patient age and sex, primaries (lung, breast, and other cancers), number, total volume, and locations of BMs, and radiotherapy were evaluated by the Cox proportional hazards regression model (Table 2). According to the multivariate analysis, the infratentorial involvement of BMs was an independent risk factor of poor prognosis for patients who received surgery (*p* = 0.023, hazard ratio = 1.473, 95% confidence interval = 1.055–2.058). In addition, we performed the ADIFFI in 200 patients, and found patients with shorter survival (≤1 year) presented significant infratentorial clusters of BMs (Appendix A).

## 3. Discussion

The current study investigated the spatial distribution of BMs via voxel-wise mapping and ADIFFI in a large cohort, and found the relationship between the infratentorial localization of BMs and specific clinical characteristics, as well as the prognostic value of infratentorial involvement.

Remarkable heterogeneity is observed in both the biological characteristics and molecular markers of BMs because of the diverse origins of BMs [17,18]. In addition, one of the clinical features that potentially reflect the heterogeneity is the radiological manifestations of BMs, especially the specific distribution of lesions, which warrants evaluation for local treatments. Due to the high sensitivity, MRI is regarded as a critical radiological method to detect BMs and perform differential diagnosis by either visual inspections or data-mining analyses [19,20]. The MRI voxel-wise analysis is one of the data-mining methods and is applied in investigating brain diseases, such as stroke and amyloid pathology [21,22]. The notable advantage of voxel-wise approach is registering the MRI to a standard brain template (e.g., MNI152) for normalization, realizing the transformation of specific voxels to corresponding coordinates, which is immune to the MRI field strength [23]. Our previous work showed the preferred locations of meningiomas using the voxel-wise analysis [24]. Therefore, the voxel-wise approach could be useful to unfold the heterogenetic localization of BMs.

Based on the voxel-wise method, the landscape of BMs was illustrated, with the significant site predilection in the cerebellum first found in BMs from lung and breast cancer [10,12]. Moreover, three subtypes of breast cancer (triple-negative, HER2 positive, and luminal cancers) showed site differences of BMs [14]. Schroeder et al. further mapped the distribution of BMs according to more primary malignancies, and then found BMs from gastrointestinal cancer favored infratentorial regions, while BMs from skin cancer and sarcoma preferred supratentorial areas with the cerebellum avoided [25]. The tumor frequency heatmaps in the present study offered similar results to those mentioned above. Furthermore, the susceptibility of cerebellum to BMs was shown by analyzing the relative metastatic risk. The infratentorial localization of BMs from gastrointestinal cancer, especially colorectal cancer, was previously described and deduced to be related to the retrograde metastasis through the Batson’s plexus [26,27]. The aversion of melanoma to cerebellum was also discovered by Rogne et al. in their surgical series, and the molecular mechanism was likely the vascular endothelial growth factor (VEGF)-independent metastasis of melanoma [11,26]. 

Given these results, the infratentorial BMs might play a distinctive role. In terms of surgery, more aggressive methods were adapted to resect infratentorial BMs due to the high risks of obstructive hydrocephalus, brainstem insults, and foramen magnum herniation [26]. From the perspective of pathophysiology, two representative hypotheses pointed out that the trapping of tumor emboli in the metastatic route and the crosstalk between ‘seed’ and ‘soil’ were generally responsible for the formation of BMs [28,29]. In addition, the differences of anatomic vascularization density, hemodynamics, and oxygen content between supratentorial and infratentorial regions might explain the divergence of BMs’ formation [11]. Specifically, infratentorial BMs presented the propensity to areas with high perfusion when compared to the supratentorial lesions [30].

We, therefore, hypothesized that there were more potential relationships between the BMs’ localization and clinical characteristics. The voxel-wise ADIFFI method, used to analyze the specific localization and lateralization of glioblastoma in terms of different molecular and genetic alterations by Ellingson et al., was further applied in the present study [15,16]. Our results successfully indicated the infratentorial localization of BMs was associated with younger, male patients, lung neuroendocrine and squamous cell carcinoma BMs, and high expression of Ki-67 of primary tumors and BMs. The prophylactic cranial irradiation (PCI) was used in patients with small cell lung cancer (SCLC) and the major type of lung neuroendocrine carcinoma, with a high risk to develop BMs [5]. Though the effects of PCI in limited or extensive stages of SCLC remained controversial, the infratentorial areas were recommended to receive PCI, according to our results [31]. Meanwhile, infratentorial clusters were observed in squamous cell lung cancer BMs, which is more malignant with a worse outcome when compared to the adenocarcinoma BMs. Thus, the results indicated the infratentorial areas were more vulnerable to tumors with aggressive behaviors, supported by the following results of Ki-67 comparisons. However, compared to the above clinical features, relatively smaller predominant clusters related to poor survival were found (Figure 5E). This possibly resulted from the combined effects including the more aggressive treatments of infratentorial BMs, the satisfactory survival of younger patients, and the non-significant role of the infratentorial involvement of BMs in survival in all patients (Figure 5C).

These findings concurrently unraveled a special role of the infratentorial localization of BMs. In addition to the infratentorial areas, we also found significant clusters of shorter survival located at the bilateral frontal lobes (Figure 5E). We speculate about two reasons. (1) Though the tolerance to mass effects is relatively good due to the large volume of the frontal lobe itself, one unilateral lesion might spread to the contralateral frontal lobe forming diffuse diseases. (2) Bilateral frontal BMs would significantly impair the neurocognitive functions.

As mentioned, no statistically significant difference was found when evaluating the effect of BMs locations on overall survival (Figure 5C). Kancharla et al. reviewed the literature and suggested the effect of BMs location on clinical outcomes was controversial, largely due to the limitations of different studies [32]. The definition of locations was one of the important problems. The anatomical identification of brain lobes and metastatic lesions that affected more than one lobe could be voxel-wise accomplished by MNI152 registration and a centroid calculation. However, different location categories of patients with multiple/diffuse BMs (e.g., supratentorial versus infratentorial, eloquent versus non-eloquent) might impact the outcome analysis. We divided patients into groups with or without infratentorial involvement. The results indicated that, for patients who received surgery, infratentorial involvement of BMs was an independent risk factor of worse outcomes. Enders et al. reported a significantly worse survival in surgically treated non-small cell lung cancer (NSCLC) patients with infratentorial BMs [33]. BMs with infratentorial or both supratentorial and infratentorial localization was proven to be an independent factor of death in a cohort investigating the dose of WBRT [34]. However, when comparing cerebellar BMs with supratentorial ones in patients received radiosurgery, there was no significant prognostic value in multivariate analysis [35]. Therefore, the prognostic value of BMs’ locations might be varied with different cohorts and location categories.

The voxel-wise analysis additionally found the differences of BMs volume according to various brain areas and primaries, while also implying the heterogeneity of BMs. Compared to the preferred distribution of infratentorial regions, the hippocampus was demonstrated to have a low risk of developing BMs [10,36]. There was a remarkably low proportion of hippocampus BMs in Figure 3. Accordingly, patients were verified to benefit from hippocampal-avoidance whole-brain radiotherapy (HA-WBRT) in avoiding unnecessary radiation toxicities to neurocognition when compared with conventional WBRT based on clinical trials [37,38]. Moreover, we found patients who only received WBRT presented the worst OS (Figure 5D). Apart from the potential neurocognitive toxicities, using WBRT alone is primarily adopted as a palliative treatment in patients with extensive BMs. Additionally, the patients who received WBRT alone had significantly more BMs (*p* < 0.001, 5.34 ± 0.52 vs. 1.56 ± 0.15, Appendix A) than those who received other therapies in our cohort, leading to an unsatisfactory OS.

There are several limitations in our study. Previous studies have reported the type of primaries was lung cancer (43.3%), followed by melanoma (16.4%), breast cancer (15.7%), colorectal cancer (9.3%), renal cell carcinoma (9.1%), and other cancers (6.2%) [39]. In our study, 86.1% of the cases were diagnosed with lung cancer BMs. Fabi et al. noted that, compared to lung cancer BMs, brain lesions emerged much later after the diagnosis of other primary tumors (e.g., 46 months for breast cancer, 42 months for colorectal cancer, and 22 months for melanoma) [4]. Global statistics showed that several cancers in eastern Asia had similar age-standardized incidence rates to northern America (e.g., 47.2 vs. 39.1 per 100,000 for male lung cancer), but there was a much lower incidence of breast and skin cancers (39.2 vs. 84.8 for breast cancer and 1.4 vs. 76.9 for male skin cancer) [40]. In addition, the single-center nature of this study also partly explains the large inclusion amount of lung cancer. Patients in recent times were more likely to be enrolled for a single-center analysis, most of whom were lung cancer BMs, while those who developed BMs at a later stage, such as breast cancer patients, might be examined in another institution, which could not be followed up. Thus, the presence of BMs varies with the biological characteristics of specific malignancies, and their different incidence rates in distinct countries and regions may result in differences in the types of primaries. We have avoided unqualified claims in our results, such as just making descriptive analyses for tumor frequency heatmaps and not performing ADIFFI between lung cancer and other cancers with a small sample. Thus, a more balanced and multi-institutional patient cohort is imperative. Secondly, most of the patients had the histological verification only of the primary tumors. The histological verification as a golden standard might be required. Thirdly, the molecular and genetic mechanisms of the infratentorial predilection of BMs were not shown. It was reported that special mutations existed in the infratentorial IDH-mutant astrocytomas and pediatric cerebellar high-grade gliomas [41,42]. Therefore, laboratory investigations were needed to further interpret our findings. Lastly, the leptomeningeal metastases are important for a poor prognosis. However, because of the limitation of the voxel-wise segmentation that we used, the leptomeningeal enhancement became an unmeasurable lesion. It is under our effort to improve the methods and further elucidate the role of leptomeningeal enhancement.

## 4. Materials and Methods

### 4.1. Patients

A total of 3130 patients initially diagnosed with BMs via contrast-enhanced T1 weighted MRI (CE-T1WI) at our institution from January 2012 to June 2017 were reviewed. Among the 3130 patients, 105 were excluded from the study because the metastatic lesions were located only in the skull or scalp. Histopathological reports for the remaining 3025 patients were extracted from a database. Matched records for 1323 patients were found and were categorized based on primary tumors and BMs biopsies. Patients who met the following criteria were included: (1) patients with a BMs biopsy, with or without a biopsy of primary malignancies, and (2) patients with only a biopsy of primary malignancies (due to surgical contraindications for BMs). Patients who met the following criteria were excluded: (1) histopathological data were inconclusive or invalid, or (2) BMs originated from the brain, or (3) patients had more than one type of primaries but lacked a BMs biopsy. A total of 1102 patients were analyzed further after strict review and differential diagnosis of MRI by a senior neuroradiologist (B.J.). Figure 1A shows a schematic of the selection procedure. The data presented in this study are available on request from the corresponding author.

### 4.2. Patient Consent

The present study was in compliance with the Declaration of Helsinki. Consent agreement that stating the radiological and histopathological data, and medical records might be used for teaching or scientific research was signed by every patient as soon as the admission to hospital. The retrospective data collection and the submission for publishing this study were approved by the Institutional Ethical Committee on clinical human research (No. 2020–801). Medical records were deidentified for privacy protection.

### 4.3. MRI

The patients included in the current study underwent a 1.5 (Signa Excite, GE Healthcare, Milwaukee, Wisconsin) or 3.0 Tesla (Discovery 750, GE Healthcare, Milwaukee, Wisconsin) MRI. Intravenous injection of gadodiamide (0.2 mL/kg body weight, up to a maximum of 20 mL, Omniscan, GE Healthcare) was used to obtain the CE-T1WI. The contrast-enhanced sequence is turbo spin echo (TSE) with a section thickness of 6 mm. The repetition time (TR) and echo time (TE) for 1.5 Tesla MRI are 400–1750 ms and 6.9–10 ms, respectively. The TR and TE for 3.0 Tesla MRI are 1750–2080 ms and 24.62–25.15 ms, respectively. With regard to the patients with multiple scans, the first CE-T1WI scan demonstrating BMs was used for analyses.

### 4.4. MRI Data Processing

The CE-T1WI were first registered to Montreal Neurological Institute 152 brain template and coordinate system (MNI152, Montreal Neurological Institute, McGill University, Montreal, Quebec, Canada) for normalization, according to the previous studies [13,43]. Briefly, images in the standard Digital Imaging and Communications in Medicine (DICOM) format were converted to the Neuroimaging Informatics Technology Initiative (NIfTI) format using dcm2nii converter software (University of Nottingham School of Psychology, Nottingham, UK). The ‘Normalize estimate and write’ module with default parameters of the Statistical Parametric Mapping Software version 12 (SPM12, Institute of Neurology, University College London, London, UK) in MATLAB (version R2012a, The MathWorks, Natick, MA, USA) was used to perform the registration (Appendix A). After that, hyperintense and ring-enhanced BMs in normalized images were semi-automatically segmented using 3D Slicer (version 4.10.0; http://www.slicer.org/) to form three-dimensional regions of interest (ROIs) [12,44]. Voxels within each lesion were drawn manually, followed by automatic filling with the ‘Grow from Seed’ module. The segmentation was performed by three authors (Z.D., J.W., and Q.Y.), who were trained to operate the software. The results of segmentation were independently reviewed by a neurosurgeon (C.S.) and a neuroradiologist (B.J.), both with more than 15 years of experience. Figure 1B presents the procedure of imaging data processing.

### 4.5. Calculation of the BMs Locations, Number, and Volume

An ROI-wise matrix calculation was conducted to acquire the three-dimensional coordinates of the tumor centroid via custom MATLAB scripts (Appendix A). The brain areas were artificially divided into the frontal, temporal, parietal, occipital and insular lobes, limbic system (defined as the structure including the cingulate gyrus, hippocampus, parahippocampal gyrus, and amygdala), thalamus and basal ganglia, cerebellum, and brainstem (the cerebellar vermis and brainstem were categorized as midline structures, and other regions were divided into left and right sections). The tumor sites were then determined by comparing the centroid with the coordinates of different areas (Appendix A).

Additionally, the number and volume (technically, the normalized volume, as the measurement is not an absolute measurement of lesion volume after normalization) of BMs were analyzed with custom MATLAB scripts (Appendix A). Two adjacent ROI masks were objectively differentiated based on the ‘bwlabeln’ function in MATLAB (i.e., a voxel-voxel connection through at least one edge of the cube of the voxels was classified as one lesion). The volume of each lesion after normalization or each predefined brain region was obtained by multiplying the voxel counts within the lesion or the region by the volume of a single voxel (0.08 mm^3^).

### 4.6. Construction of Tumor Frequency and p-Value Heatmaps

The ROIs were registered to the MNI152 coordinate system and overlapped using MRIcron (University of Nottingham School of Psychology, Nottingham, UK) to create tumor frequency heatmaps (Figure 1B) [13]. Then, the *p*-value heatmaps comparing two differential phenotypes under one characteristic (e.g., older and younger patients stratified by age) were constructed. Two tumor frequency heatmaps of phenotype A and B were first constructed, respectively. Then a 2 × 2 contingency table was used to perform a two-tailed Fisher’s exact test to realize the analysis of differential involvement (ADIFFI), as previously described by Ellingson et al. [15,16]. According to Fisher’s exact test, the probability of obtaining an observed pattern was calculated.
(1)$$p=\frac{\left(a+b\right)!\left(c+d\right)!\left(a+c\right)!\left(b+d\right)!}{a!b!c!d!n!}$$

This formula was used to calculate the significance of a particular voxel, where ‘a’ is the frequency of tumor occurrence under phenotype A, ‘b’ is the frequency of tumor occurrence under phenotype B, ‘c’ is the frequency of tumor-free patients under phenotype A, ‘d’ is the frequency of tumor-free patients under phenotype B, and ‘n’ is the total number of patients (Appendix A).

### 4.7. Normalization of the Metastatic Risk of BMs

The relative metastatic risk (RMR) is a measurement of the likelihood of BMs occurrence (not considering primaries types). The RMR in a specific brain region was normalized to the volume of the region in order to diminish the impact of non-uniform volume of different brain regions.
(2)$${\mathrm{RMR}}_{x}=\frac{N_{x}{/N}_{p}/V_{x}}{\sum_{i=1}^{{n=N}_{R}}{(N}_{i}/N_{p}/V_{i})/N_{R}}$$

N_x_ is the number of patients with BMs in a specific brain region. N_p_ is the sample size (1102 in our study). V_x_ is the volume of a specific brain region in MNI152, and N_R_ is the number of brain regions (18 in our study). After normalization, RMR_x_ represents the relative metastatic risk of BMs in the per unit volume of a specific brain region. RMR_x_ > 1 indicates BMs susceptibility. RMR_x_ = 1 indicates BMs neutrality, and RMR_x_ < 1 indicates BMs non-susceptibility.

### 4.8. Statistical Analyses

All data were presented as mean ± standard error of the mean (SEM). Data were analyzed using the Mann-Whitney U or Kruskal-Wallis tests. Dunn’s multiple comparison test was used for post hoc pairwise comparisons. The follow-up data were available in 991 patients, and OS was defined as the time from the first radiological detection of BMs until death or the last follow-up. Clinical features of BMs were used to perform the Kaplan-Meier analysis with the log-rank test. The Cox proportional hazards regression models were constructed to eliminate compounding factors and identify independent prognostic factors. *p* < 0.05 was considered statistically significant. All statistical analyses were performed in GraphPad Prism (version 8.0.2, GraphPad Software, San Diego, CA, USA) and SPSS (version 22.0, IBM SPSS Statistics, Armonk, NY, USA).

## 5. Conclusions

In conclusion, the current study suggested the infratentorial localization of brain metastases might correlate with specific clinical characteristics and portend worse outcomes by voxel-wise analysis. It may add valuable clinical recognition of BMs and provide references for BMs diagnosis, treatment evaluation, and prognostic prediction.

## Figures and Tables

**Figure 1 cancers-13-00324-f001:**
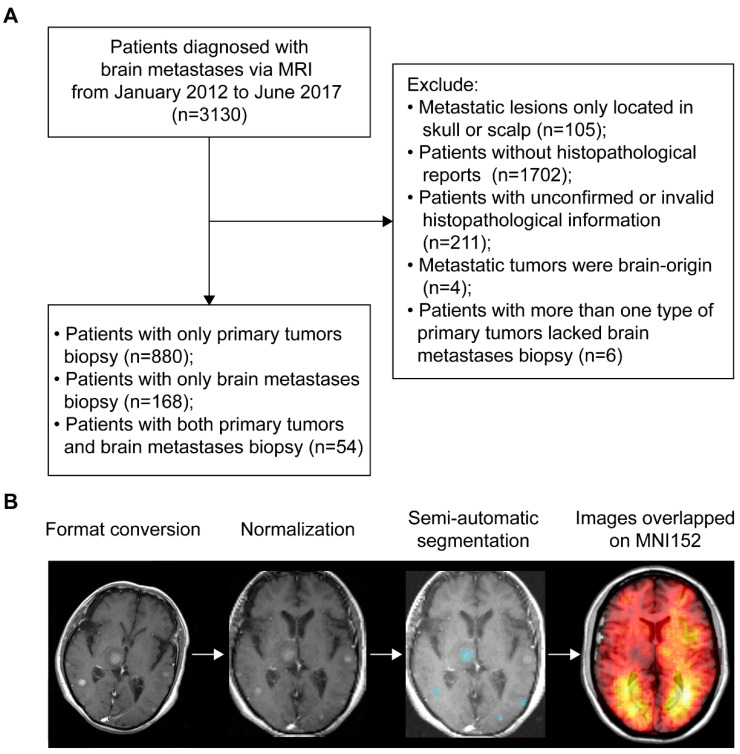
Patients inclusion and MRI data processing. (**A**) The workflow of inclusion and exclusion of brain metastases (BMs) patients. (**B**) Schematics of MRI data processing. The blue areas represent the regions of interest. MNI152 refers to the Montreal Neurological Institute 152 brain template and coordinate system. MRI refers to magnetic resonance imaging.

**Figure 2 cancers-13-00324-f002:**
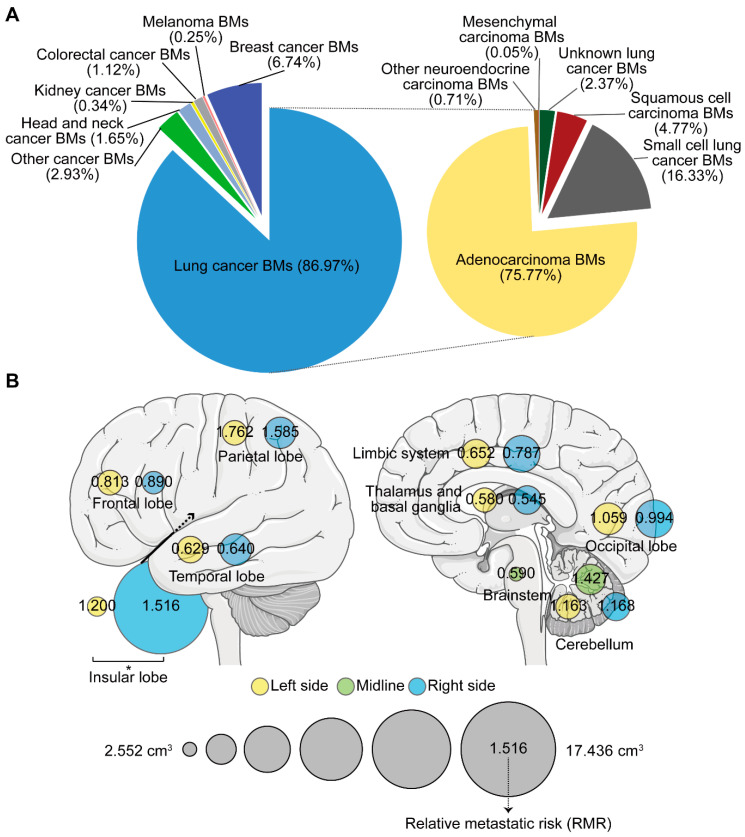
The number, volume, and relative metastatic risk of brain metastases (BMs). (**A**) Composite pie charts indicate the percentage of BMs according to different primary malignancies. The right chart represents the percentage of BMs from different subtypes of lung cancer. (**B**) Schematics showing the average volume of the single lesion of BMs on different sides of distinct brain lobes (circle size), and the corresponding relative metastatic risk (RMR) after normalization to the volume of the brain lobe. BMs refer to brain metastases. *: *p* < 0.05.

**Figure 3 cancers-13-00324-f003:**
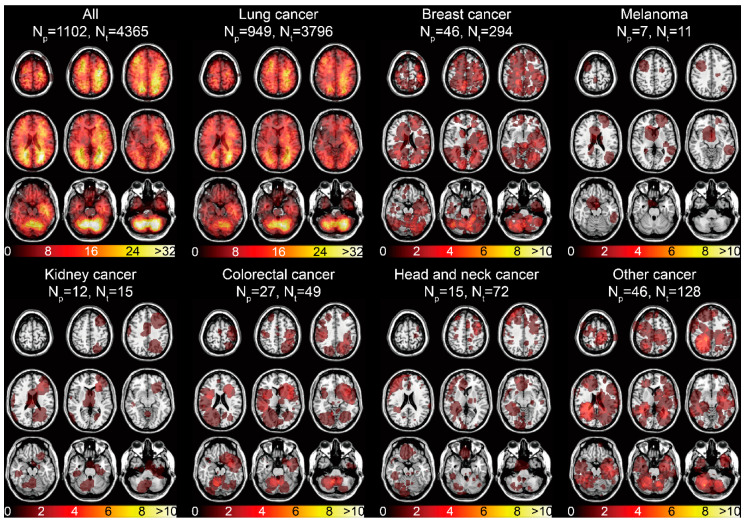
Tumor frequency heatmaps of BMs according to different primary malignancies. The numbers in the color bars refer to the number of BMs. N_p_ refers to the number of patients. N_t_ refers to the number of tumors.

**Figure 4 cancers-13-00324-f004:**
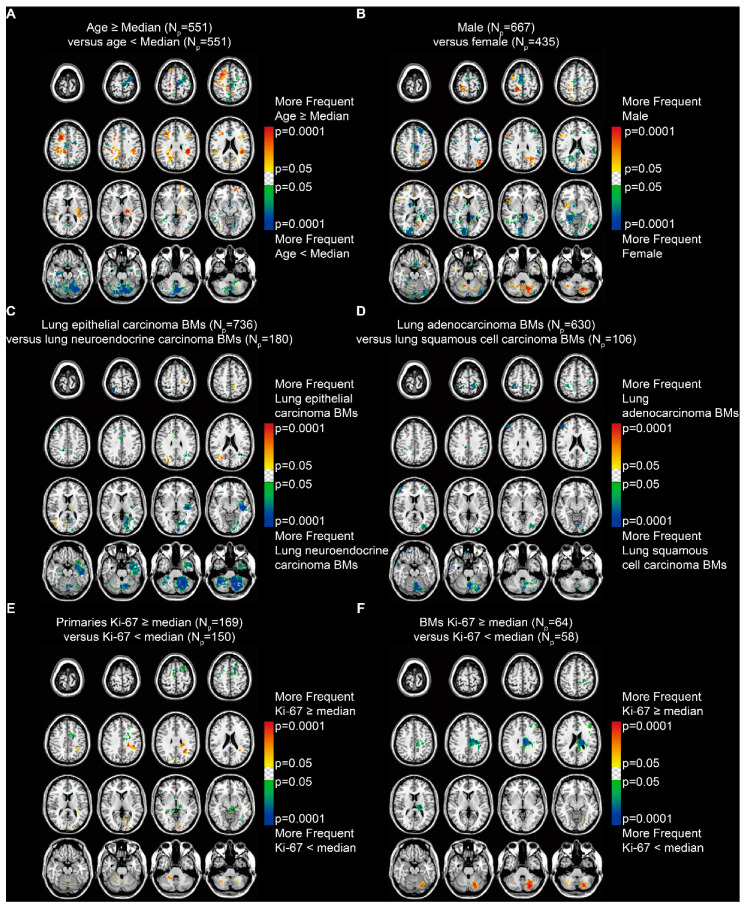
*p*-value heatmaps constructed by analysis of differential involvement (ADIFFI) displayed BMs’ cluster predominance. (**A**) Heatmaps comparing patients ≥ the median age (60.5 years) with those < the median age. (**B**) Heatmaps comparing males with females. (**C**) Heatmaps comparing patients with lung epithelial carcinoma BMs with those with lung neuroendocrine carcinoma BMs. (**D**) Heatmaps comparing patients with lung adenocarcinoma BMs with those with lung squamous cell carcinoma BMs. (**E**) Heatmaps comparing patients with Ki-67 positivity in primaries ≥ the median (35%) with those < the median. (**F**) Heatmaps comparing patients with Ki-67 positivity in BMs ≥ the median (45%) with those < the median. BMs refer to brain metastases. N_p_ refers to the number of patients.

**Figure 5 cancers-13-00324-f005:**
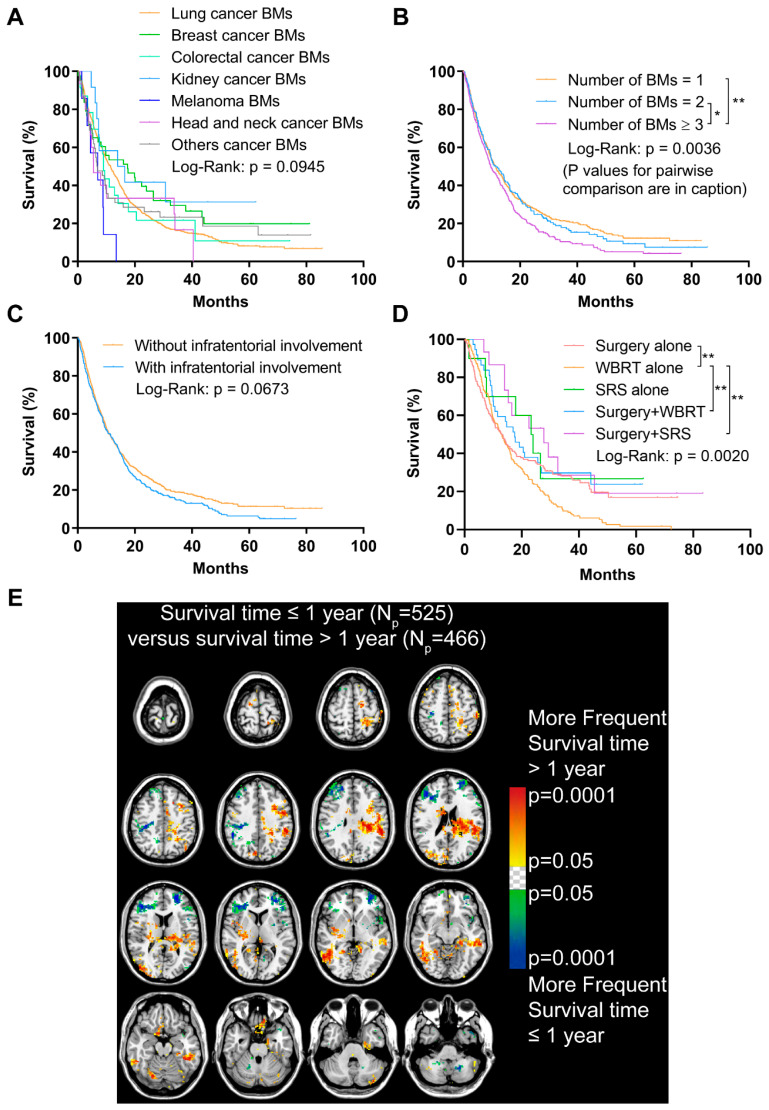
Analyses of the overall survival (OS) and analysis of differential involvement (ADIFFI) of survival-related localization of BMs. (**A**) OS analysis comparing patients with BMs originated from different primary malignancies. (**B**) OS analysis comparing patients with 1, 2, and ≥3 BMs (*n* = 1 vs. *n* = 2, *p* = 0.605; *n* = 1 vs. *n* ≥ 3, *p* = 0.0010; *n* = 2 vs. *n* ≥ 3, *p* = 0.0347). (**C**) OS analysis comparing patients with or without infratentorial involvement of BMs. (**D**) OS analysis comparing patients who received different treatments (surgery alone vs. WBRT alone, *p* = 0.005; surgery plus WBRT vs. WBRT alone, *p* = 0.006; surgery plus SRS vs. WBRT alone, *p* = 0.005). (**E**) *p*-value heatmap constructed by ADIFFI displayed BMs cluster predominance when comparing patients with OS ≤1 year with OS >1 year. BMs refer to brain metastases. N_p_ refers to the number of patients. WBRT refers to whole-brain radiotherapy. SRS refers to stereotactic radiosurgery. *: *p* < 0.05 and **: *p* < 0.01.

**Table 1 cancers-13-00324-t001:** Demographics.

Parameter	Category	Number (%)
Age at diagnosis (years)	Range	16–89
	Average	59.8
Sex	Male	667 (60.5)
	Female	435 (39.5)
Primary malignancies	Lung cancer	949 (86.1)
	Adenocarcinoma	630 (66.4)
	Squamous cell carcinoma	106 (11.2)
	Small cell lung cancer	169 (17.8)
	Other neuroendocrine subtypes	11 (1.1)
	Mesenchymal neoplasms	2 (0.2)
	Others	31 (3.3)
	Breast cancer	46 (4.2)
	Melanoma	7 (0.6)
	Kidney cancer	12 (1.1)
	Colorectal cancer	27 (2.5)
	Head and neck cancer	15 (1.3)
	Other cancer	46 (4.2)
Brain metastases number	N = 1	544 (49.4)
	N = 2	188 (17.0)
	N ≥ 3	370 (33.6)
Overall survival for 991 patients (months)	Range	0.10–85.55
	Median survival	10.84
	95% Confidence Interval	9.768–11.915
Treatments	Surgery alone	146 (36.3)
	Whole-brain radiotherapy alone	194 (48.3)
	Stereotactic radiosurgery alone	10 (2.5)
	Surgery + whole-brain radiotherapy	37 (9.2)
	Surgery + stereotactic radiosurgery	15 (3.7)

**Table 2 cancers-13-00324-t002:** Cox proportional hazards regression for brain metastases patients who received surgery.

Factors	Number	Univariate		Multivariate	
		HR (95% CI)	*p* Value	HR (95% CI)	*p* Value
Age					
(Median = 59.1 years)					
Young	100	Ref	Ref	Ref	Ref
Old	100	0.883 (0.640–1.218)	0.449	0.892 (0.643–1.238)	0.496
Sex					
Female	78	Ref	Ref	Ref	Ref
Male	122	1.293 (0.928–1.802)	0.129	1.239 (0.857–1.790)	0.255
Primaries					
Lung cancer	138	Ref	Ref	Ref	Ref
Breast cancer	14	1.077 (0.576–2.015)	0.816	1.209 (0.610–2.398)	0.586
Other cancer	48	1.128 (0.772–1.650)	0.533	1.150 (0.776–1.705)	0.486
Number					
N = 1	161	Ref	Ref	Ref	Ref
N = 2	25	1.142 (0.710–1.835)	0.584	1.036 (0.633–1.697)	0.888
N ≥ 3	14	2.132 (1.198–3.793)	0.010 *	1.751 (0.944–3.245)	0.075
Total volume					
(Median = 27.67 cm^3^)					
Small	100	Ref	Ref	Ref	Ref
Large	100	1.325 (0.961–1.828)	0.086	1.245 (0.897–1.728)	0.190
Locations					
Without infratentorial involvement	136	Ref	Ref	Ref	Ref
With infratentorial involvement	64	1.473 (1.055–2.058)	0.023 *	1.473 (1.055–2.058)	0.023 *
Radiotherapy ^1^					
No	146	Ref	Ref	Ref	Ref
Yes	54	0.738 (0.512–1.064)	0.103	0.778 (0.535–1.129)	0.186

^1^ Radiotherapy includes whole-brain radiotherapy and stereotactic radiosurgery. * *p* values with statistical significance. Ref refers to reference. HR refers to hazard ratio. CI refers to confidence interval.

## Data Availability

The data presented in this study are available on request from the corresponding author.

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
