# Peer review of "The Infratentorial Localization of Brain Metastases May Correlate with Specific Clinical Characteristics and Portend Worse Outcomes Based on Voxel-Wise Mapping"

_cancers, 2021, doi:10.3390/cancers13020324_

Round 1

Reviewer 1 Report

Authors should provide enough empirical evidence in results to supports the claims made in paper

Reviewer 2 Report

The authors present an interesting analysis of brain metastasis location as it pertains to primary tumor type, clinical characteristics, and prognosis. The paper is well put together and includes an impressively large dataset with >1000 patients and >4000 lesions. I think this is a nice study. However, I do think some more methodological detail should be included in the methods section to more precisely describe what the authors have done. I also think some more attention should be paid to the large class imbalance amongst primary tumor types. The vast majority of cases were lung primary, so any claims about the other subgroups (i.e. sec 2.3) should be heavily qualified.

General comments:

  • I assume the ordering of the sections (i.e. results, discussion before methods) comes from a prior submission to another journal. I would suggest re-ordering for the sake of clarity. But that is ultimately an editorial decision.
  • More detail is needed in the methods section (xspecific requests are below). I find it slightly unclear exactly what was done.
  • There is of course a link between BM location and surgical operability (as the authors do discuss). It would be interesting to see a BM probability heatmap as a function of survival (like in Fig 5E) with a control for intervention type. I should note that the authors do show (sec 2.6) that infratentorial involvement is predictive of survival independent of surgery.
  • There is a huge imbalance in primary tumor type amongst the BM cases here. The authors should at least comment on how this affects statistical findings regarding differences between tumor groups.
  • Perhaps this is the plan post-review, but it would probably be more useful to provide the matlab code in .m format rather than typed into a pdf or word document.

Simple summary:

Final sentence: I think it should read “prognostic prediction of BMs”

Methods:

P13 L329: “Medical records were acquired and desensitized for privacy protection”. I would suggest saying they were “deidentified”

P13 L363: why is the lesion volume “technically the normalized volume?” please explain.

P14 L370: in section 4.4 you state that SPM is used to register to MNI152. Please clarify.

P14 Sec4.6: I’m a bit confused by this section. The formula you show only discussed two characteristics yet you have numerous variables (primary tumor type + clinical characteristics). Are multiple head to head comparisons performed? (based on the results section I believe this is the case, but it should be described here). Also, are the probabilities of observing a tumor given phenotype A, e.g. computed using the ”tumor frequency heatmaps”?

P14 Sec 4.7: I am similarly confused as in 4.6. All you describe is a probability of BM vs. no BM in a given region. Are you performing separate calculations for each subgroup of patients (i.e. divided by primary tumor type)?

P14 L389: what is meant by “normalization” of the RMR?

P14 :391: What do you mean by BM susceptibility and resistance? That BMs are more/less likely to appear in a given region for a given primary tumor type? That BMs are more/less likely to appear in a given region than in other regions?

Results:

Figure 2: I don’t quite understand what the sizes mean. Is that the average BM size in that region?

Reviewer 3 Report

This paper presents interesting results on the correlation between infratentorial localization of brain metastases and some clinical characteristics. In general, the method used in the paper seems sound, and shows interesting results. In my opinion, the paper is suitable for publication. I have only a minor remark:

 Fig.1. Is in not clear the “Images Superimposition”.  Super imposed on what? Some details reported in the ‘Materials and Methods’ Section should be reported here.

Author Response

We appreciate your comments. We considered the word 'superimposition' might not be suitable and clear. After acquiring the ROIs, they were overlapped on the brain template MNI152 to obtain the tumor frequency heatmap. As a result, we replaced the 'superimposition' by using 'overlap', and the text and Figure 1 was correspondingly revised (Figure 1, line 141, 418).

Reviewer 4 Report

In this retrospective study, authors investigated the potential relationship between infratentorial localization of brain metastases from different histotypes and clinical characteristics including age, sex, primaries, Ki-67 index, and overall survival. With this purpose they evaluated 1102 patients with 4365 brain metastases applying a Voxel-wise mapping of MRI to construct the tumor frequency heatmaps after normalization and segmentation. The results confirmed the distinctive role of infratentorial localization of metastases and its potential relationship with specific clinical characteristics, which may assist diagnosis, treatment choice, and prognosis prediction.

- I suggest modifying the structure of the manuscript by using the following order of paragraphs: Introduction, Materials and Methods, Results, Discussion and Conclusions

Methods:

- MRI: What kind of T1 post contrast sequences were used for the study? TSE or Gradient echo? Detailed information of the sequences utilized with both 1.5 and 3.0T units are required to have a better idea of homogeneity of data.

- Statistical analysis: line 396 “ Long term follow-up data”. What are the authors referring specifically to? How many follow-up? How far distant in time they go? Has been the appearance of new lesions being considered? Is the association with leptomeningeal enhancement being considered and assessed?

Results:

- Demographics: line 84 “were confirmed”, better to substitute with “were selected”

- Demographics: Most of the patients included had histological verification only of the primary tumors. What was done in term of differential diagnosis? Which were the criteria used to rule out alternative diagnoses such as abscess? No mention of alternative diagnosis for the non-histologically proven met is made across the paper and particularly among the limitation.

- Lines 184-185: “no significance was found between” what do the author mean with this statement? Do they mean that no statistically significant difference in OS was found between… ? if yes this should be better expressed in English. If not, authors should better explain what they meant. See also line 273 of the discussion.

- Lines 186-188: “Additionally, a significantly worse OS was observed in patients who only received WBRT…”. How do the authors explain this obervation? Is this because patient who get WBRT have more spread intracranial metastatic disease?

- 2.6: Was an independent risk factor for what? Does this refer to the risk of developing a brain met within the infratentorial region after surgery elsewhere in the brain? If yes this should be better explained. The nature and meaning of this analysis is not completely understood. See also discussion line 288-289. This statement is very generic and does not get to the point of what the finding means

-  Lines 192-193: This is an interesting finding, is there any hypothesis for explaining it? In the discussion I see some hypothesis regarding the involvement of the infratentorial region but nothing about the bilateral frontal lobe involvement.

- Figure 2A: Change the color of the text: “Adenocarcinoma BM” in the circle in the upper right since it is not readable.

Round 2

Reviewer 1 Report

Authors has resolved the previous comments in its revised manuscript.

Reviewer 4 Report

The authors have invested a satisfactory amount of work to respond to the reviewers comments and to justify their methodology and results obtained in the reaction letter.

The quality of the manuscript is now significantly improved and I am substantially satisfied.